# The Impact of Melatonin Supplementation and NLRP3 Inflammasome Deletion on Age-Accompanied Cardiac Damage

**DOI:** 10.3390/antiox10081269

**Published:** 2021-08-10

**Authors:** Ramy K. A. Sayed, Marisol Fernández-Ortiz, Ibtissem Rahim, José Fernández-Martínez, Paula Aranda-Martínez, Iryna Rusanova, Laura Martínez-Ruiz, Reem M. Alsaadawy, Germaine Escames, Darío Acuña-Castroviejo

**Affiliations:** 1Centro de Investigación Biomédica, Departamento de Fisiología, Facultad de Medicina, Instituto de Biotecnología, Parque Tecnológico de Ciencias de la Salud, Universidad de Granada, 18016 Granada, Spain; ramy.kamal@vet.sohag.edu.eg (R.K.A.S.); sol92@correo.ugr.es (M.F.-O.); josefermar@ugr.es (J.F.-M.); ampaula@correo.ugr.es (P.A.-M.); irusanova@ugr.es (I.R.); lauramartinezruiz8@gmail.com (L.M.-R.); gescames@ugr.es (G.E.); 2Department of Anatomy and Embryology, Faculty of Veterinary Medicine, Sohag University, Sohag 82524, Egypt; 3Département de Biologie et Physiologie Cellulaire, Faculté des Sciences de la Nature et de la Vie, Université Blida 1, Blida 09000, Algeria; rahim.im15@gmail.com; 4CIBERfes, Ibs.Granada, 18016 Granada, Spain; 5Department of Zoonoses, Faculty of Veterinary Medicine, Assiut University, Assiut 71526, Egypt; reem.barbary@vet.au.edu.eg; 6UGC de Laboratorios Clínicos, Hospital Universitario San Cecilio, 18016 Granada, Spain

**Keywords:** cardiomyocytes, NLRP3 inflammasome, mitochondria, CSA, ultrastructure, sarcopenia, melatonin, autophagosome, β-MHC

## Abstract

To investigate the role of NLRP3 inflammasome in cardiac aging, we evaluate here morphological and ultrastructural age-related changes of cardiac muscles fibers in wild-type and NLRP3-knockout mice, as well as studying the beneficial effect of melatonin therapy. The results clarified the beginning of the cardiac sarcopenia at the age of 12 months, with hypertrophy of cardiac myocytes, increased expression of β-MHC, appearance of small necrotic fibers, decline of cadiomyocyte number, destruction of mitochondrial cristae, appearance of small-sized residual bodies, and increased apoptotic nuclei ratio. These changes were progressed in the cardiac myocytes of 24 old mice, accompanied by excessive collagen deposition, higher expressions of IL-1α, IL-6, and TNFα, complete mitochondrial vacuolation and damage, myofibrils disorganization, multivesicular bodies formation, and nuclear fragmentation. Interestingly, cardiac myocytes of NLRP3^−/−^ mice showed less detectable age-related changes compared with WT mice. Oral melatonin therapy preserved the normal cardiomyocytes structure, restored cardiomyocytes number, and reduced β-MHC expression of cardiac hypertrophy. In addition, melatonin recovered mitochondrial architecture, reduced apoptosis and multivesicular bodies’ formation, and decreased expressions of β-MHC, IL-1α, and IL-6. Fewer cardiac sarcopenic changes and highly remarkable protective effects of melatonin treatment detected in aged cardiomyocytes of NLRP3^−/−^ mice compared with aged WT animals, confirming implication of the NLRP3 inflammasome in cardiac aging. Thus, NLRP3 suppression and melatonin therapy may be therapeutic approaches for age-related cardiac sarcopenia.

## 1. Introduction

Aging is a complex process that depends on diverse factors including environmental, cellular, and genetic ones, resulting in accumulation of cellular and molecular defects, and finally displaying dysfunctional impacts on tissues, organs, and systems [1]. Age is also accompanied with morphological and biochemical changes, yielding a progressive and irreversible decline of all functions of the body [2].

Cardiovascular diseases comprise the master cause of death worldwide [3], where many risk factors have been identified, including obesity, hypertension, smoking, genetics, lipid profile, insulin resistance, and physical inactivity [4]. During aging, progressive impairments of diverse metabolic pathways were reported including inflammation and mitochondrial and autophagy dysfunction [5]; a considerable amount of these changes are implicated in cardiac aging and age-accompanied cardiovascular diseases [6].

With mammalian aging, muscle mass and strength are progressively lost, a phenomenon defined as sarcopenia, which is characterized by a set of biochemical and morphological changes that deteriorate muscle function [7]. Sarcopenia has also been considered one of the most essential reasons for reduced physical performance and declined cardiorespiratory fitness in older individuals with heart failure [8].

Clinical and experimental studies have been suggested that aging is accompanied by significant modulations in cardiac structure and function, including several histopathologic changes in mouse hearts with age [9]. Among these, subendocardial and interstitial fibrosis, variable and hypertrophic myocyte fiber size, vacuolization of cytoplasm, collapse of sarcomeres, mineralization, and arteriolosclerosis were observed [10]. Furthermore, age-associated cardiomyocytes hypertrophy, increased cardiomyocyte apoptosis, and fibrosis were more commonly observed in the ventricular subendocardium of the old mouse hearts [11].

Inflammation has been proposed as a risk factor associated with cardiovascular diseases [12]. Inflammaging is a chronic low-grade inflammation, which develops with aging and describes age-accompanied inflammatory response upregulation [13]. NLRP3 inflammasome is upregulated after several cardiovascular diseases such as atherosclerosis, myocardial infarction, hypertension, chronic heart failure, ischemic heart disease, and diabetic cardiomyopathy [12,14]. Recent studies reported the role of the NLRP3 (nucleotide-binding domain, leucine-rich containing family, pyrin domain-containing-3) inflammasome in cardiovascular diseases and cardiac aging. Lack of NLRP3 in mice has been reported to improve several age-associated degenerative changes with inducing muscle strength and minimizing age-related myopathic fibers [15].

Activation of the NLRP3 inflammasome requires two signals. The first signal is the microbial molecules or endogenous cytokines, which upregulate the expression of NLRP3 and pro-inflammatory cytokine interleukin-1β (pro-IL-1β) via nuclear factor kappa B (NF-κB) pathway activation [16]. The second signal is stimulated by different damage-associated molecular patterns, resulting in formation of the inflammasome multiprotein complex. The inflammasome activation induces through caspase-1 activation of pro-IL-1β and pro-IL-18 into their mature process and enhances their subsequent secretion. Finally, inflammatory cytokine interleukin-1β (IL-1β) and inflammatory cytokine interleukin-18 (IL-18) initiate an inflammatory process of regulated cell death [17,18]. Furthermore, age-related reduction of the endogenous antioxidant defense capacity including transcription factor nuclear factor erythroid 2–related factor 2 (Nrf2) decline contributes to NLRP3 inflammasome activation [19,20].

Melatonin (N-acetyl-5-methoxytryptamin, aMT) is a hormone that is synthesized in the pineal gland and is also vastly produced by various body organs and tissues [21]. Besides the chronobiotic properties of pineal melatonin, the extrapineal melatonin exerts a regulatory role in autophagy, cell cycle and apoptosis, mitochondria homeostasis, and acts as an immunomodulatory, free scavenger, and cytoprotective agent with anti-inflammatory and anti-oxidative features [22,23,24,25,26,27]. Melatonin administration improves muscular function, reduces oxidative stress, and inflammation in athletes [28]. During aging, melatonin therapy also ameliorates the function of muscular mitochondrial integrity, retrieves cardiac mitochondria from several defects [29], and restores its structural architecture [30,31]. Recently, the relation between melatonin secretion and muscular strength during aging, in addition to its role against age-related cardiovascular diseases, has been reported [32,33].

Melatonin reveals anti-inflammatory properties through inhibiting the activation of NF-κB pathway, enhancing the conservation of the mitochondrial homeostasis, and reducing the production of reactive oxygen species (ROS) and pro-inflammatory cytokine [34,35,36]. Melatonin also blunts NF-κB/NLRP3 inflammasome connection and activation during cardiac aging [16]. Our recent study showed the role of ROS in cardiac aging and revealed the connection between melatonin, Nrf2, and NLRP3 in cardiac mitochondria during aging, where lack of NLRP3 inflammasome and melatonin supplementation prevented age-associated mitochondrial dynamics changes in cardiac myocytes. Furthermore, NLRP3 inflammasome deficiency affected Bax/Bcl2 ratio, with fewer impacts on cardiac autophagy with aging compared with wild-type animals. This study reported the ability of melatonin in restoring aged-related Nrf2-dependent antioxidant capacity [37]. With these data, we consider it of interest to analyze the role of NLRP3 inflammasome in age-related morphological and ultrastructural changes of the heart that accompany age-dependent mitochondrial dynamics changes in cardiomyocytes and whether melatonin supplementation may exert an additive beneficial effect to prevent these alterations. To achieve that, left ventricles of wild-type and NLRP3-knockout mice were dissected and processed for further analyses.

## 2. Materials and Methods

### 2.1. Experimental Animals

Female NLRP3-knockout mice NLRP3^−/−^ (B6.129S6-Nlrp3^tm1Bhk^/J) on the wild-type C57BL/6J background (>10 backcrosses) were purchased from The Jackson Laboratory, (Bar Harbor, ME, USA). Mutant mice were bred to wild-type one in accordance with The Jackson Laboratory instructions. Mice were kept in the facility of Granada University under a specific pathogen-free barrier, with a striped temperature (22 ± 1 °C), 12:12 h light: dark cycle (lights on at 08:00 a.m.) and unlimited access to food and water. Some reports reported that C57/BL6 mice could be considered melatonin-deficient mice [38,39]; however, we and others detected that they produce melatonin from pineal and extrapineal tissues. Furthermore, C57/BL6 mice have been used in other experiments and showed a good response to melatonin therapy [40,41]. Therefore, this mice strain is suitable for the purpose of this study.

All experiments were approved by the University of Granada and conducted according to the Ethical Committee of Junta de Andalucía, Spain (no. 05/07/2016/130); the Spanish Protection Guide for Animal Experimentation (R.D. 53/2013), and the European Convention for the Protection of Vertebrate Animals used for Experimental and Other Scientific Purposes (CETS # 123).

Wild-type mice and NLRP3^−/−^ ones were divided into 5 groups, with 10 animals in each group. Experimental groups of mice were mature young (Y, 3 months), early-aged (EA, 12 months), early-aged treated with melatonin (EA + aMT, 12 months), old-aged (OA, 24 months), and old-aged supplemented with melatonin (OA + aMT, 24 months) mice. The Y, EA, and OA groups of animals were fed with normal rodent chow, while others received the same rodent chow containing melatonin in a dose that allows a daily intake of 10 mg/kg b.w./mouse for 2 months before killing of mice. The integration of melatonin to the chow’s pellets was done in the facility of Diet Production Unit of the Granada University. Mice of each group were weighted and separated into 2 cages with 5 mice in each. The dose of 10 mg/kg/melatonin used here has shown significant anti-aging beneficial effects in SAM mice [29,42]. Moreover, we previously detected that the therapeutical doses of the melatonin range from 5 mg to 50 mg/kg bw [43]. Here, and with guide of the human equivalent dose formula, we calculated that 10 mg/kg in rats correspond to 1.62 mg/kg for an adult of 60 kg bw, which is about 97.3 mg/d. Furthermore, doses of 50 mg to 300 mg/d have been recommended in some clinical studies, demonstrating significant benefits, with absence of side effects [44,45,46]. Consequently, a clinical trial with 50–100 mg/d melatonin in sarcopenic patients should be recommended.

### 2.2. Magnetic Resonance Imaging (MRI)

The magnetic resonance experiments were performed on a small-animal horizontal 7 Tesla USR Bruker BioSpec TM 70/20 USR magnet (Ettlingen, Germany). Mice were anesthetized with 1.5% isoflurane in air before imaging, and the breathing rate was monitored using an air balloon put on top of the lungs (SA Instruments, Inc., Stony Brook, NY, USA). The respiration rates between animals were similar for every experiment. For heart imaging, the animals were placed in prone and supine positions, with placement of non-magnetic metallic or carbon-fiber ECG electrodes on the front paw and limbs. Coronal CINE was acquired using a CINE sequence with the following parameters: TE = 1.6 ms, TR = 8 ms, number of averages = 1, flip angle = 15.0, slice thickness = 0.8 mm, image size = 192 × 192, field-of-view = 25 × 25 mm^2^. Analyses of heart length, left ventricular lumen length, and left ventricular wall thickness were applied on the acquired images.

### 2.3. Tissue Preparation for Histological Examination

For histological analysis, five mice from each group were weighted and anaesthetized by intraperitoneal injection of equithesin (1 mL/kg). After loss of all reflexes, animals were transcardially perfused with warm saline followed by trump’s fixative (3.7% formaldehyde plus 1% glutaraldehyde in saline buffer). The heart was carefully dissected, weighted after removal of excessive connective tissues, and was fixed in the trump’s fixative. Part of the left ventricle (LV) was transferred to bouin’s fixative solution for further histological analysis, while the other part was immersed in trump’s fixative for further transmission electron microscopy examination.

After proper fixation, samples of the LV were passed in ethanol 70% for washing, followed by ethanol ascending graded concentrations for dehydration, xylene for clearing, and then samples were embedded in the paraffin wax. Sections of 4 μm-thick were cut by a SLEE Mainz Cut 5062 microtome, dewaxed in xylene, rehydrated in an ethanol descending series, washed with distilled water, and were stained with Hematoxylin and Eosin (H&E) stain for general histological analysis and Van Gieson stain for differentiation of connective tissue and cardiac muscle fibers. Sections were dehydrated in an ethanol ascending series, cleared in xylene, and mounted with DPX.

The sections were examined by a Carl Zeiss Primo Star Optic microscope, and digital images were acquired using a Magnifier AxioCamICc3 digital camera (BioSciences, Jena, Germany).

### 2.4. Transmission Electron Microscopy (TEM)

Small pieces from the LV of experimental groups were fixed in a 2.5% glutaraldehyde in 0.1 M cacodylate buffer (pH 7.4) and post fixed in 0.1 M cacodylate buffer-containing 1% osmium tetraoxide with 1% potassium ferrocyanide for 1 h. Specimens were then immersed on 0.15% tannic acid for just 50 s, incubated in 1% uranyl acetate for 1.5 h, dehydrated in ethanol, and embedded in resin. Ultrathin sections (65 nm-thick) were cut by a Reichert-Jung Ultracut E ultramicrotome, stained with uranyl acetate and lead citrate, and finally examined on a Carl Zeiss Leo 906E electron microscope.

### 2.5. Fluorescent Detection of Apoptotic Nuclei

For analysis of nuclear apoptosis in the heart, paraffin sections of 4 μm-thicknesses were put in xylene for removing paraffin and were immersed in a descending series of ethanol for dehydration. Sections were washed by distilled water and dried in air, rinsed in PBS 1X, and stained with 33258 Hoechst fluorescent dye (H6024, Sigma-Aldrich, Madrid, Spain). After staining, sections examined with LEICA DM5500B fluorescent microscope, and the acquired images were used for detection of the percentage of apoptotic nuclei by two operators, in a double-blind operation, comparing obtained results subsequently. Hoechst dye is a fluorescent dye that penetrates the cellular nucleus and binds to DNA, facilitating apoptotic nuclei detection. Under a 350-nm wavelength light, this dye exhales blue fluorescent light that enables nuclear DNA visibility and nuclear fragmentation or chromatin condensation observation.

### 2.6. Morphometrical Analysis

Morphometrical analysis of cardiac myocytes number (per 100 µm^2^) and cross-sectional area (CSA), as well as the percentage of the fibrotic area (total red positive area/total field area) were performed using images of Van Gieson-stained paraffin cross sections (10 images, 40x objective, per animal). Meanwhile, intermyofibrillar (IMF) mitochondrial number (per 5 µm^2^) and CSA were detected on electron micrographs (5 images, 10,000x objective, per animal). All these measurements were analyzed by two double-blinded investigators using Image J processing software and were represented as a percentage compared with the young group.

### 2.7. Real-Time Reverse Transcription Polymerase Chain Reaction (RT-PCR)

RNA was extracted from frozen mouse hearts (five animals from each group) with the NZY Total RNA Isolation kit (Nzytech gene and expression, Lisbon, Portugal) for analysis the expression of beta-myosin heavy chain (β-MHC, hypertrophy-associated gene) and Interleukin 1α (IL-1α), Interleukin 6 (IL-6) and Tumour necrosis factor α (TNFα) (inflammatory cytokines). A preliminary proteinase K digestion step was accomplished (20 mg/mL proteinase K, 600 mAU/mL) (Qiagen, Hilden, Germany). RNA was reverse transcribed to cDNA with qScriptTM cDNA SuperMix kit (Quanta Biosciences, Gaithersburg, MD, USA). Amplification was achieved by quantitative real-time polymerase chain reaction (RT-PCR) in a Stratagene Mx3005P QPCR System (Agilent Technologies, Madrid, Spain) using SYBR^®^ Premix Ex TaqTM (Takara Bio Europe, Saint-Germain-en-Laye, France). Primer sequences (Appendix A) were designed using the Beacon Designer software (Premier Biosoft Inc., Palo Alto, CA, USA). Thermal profile of RT-PCR was as follows: 10 min at 96 °C (1 thermal cycle), and 15 s at 95 °C and 1 min at 55 °C (40 thermal cycles). Data were analyzed according to the standard curves of cDNA. Beta-actin housekeeping gene was used as an endogenous reference gene. The negative control was a template-free sample (water). The 3 months-old wild type mice were used as a calibrator sample.

### 2.8. Statistical Analysis

All the statistical analyses were performed using Prism 6 software package (GraphPad, La Jolla, CA, USA) and the data presented as means ± SEM of *n* = 5 animals per group. For statistical comparisons, one-way ANOVA with a Tukey’s post hoc test was used to compare the differences between the experimental groups. Differences were considered significant when *p* < 0.05.

## 3. Results

### 3.1. NLRP3 Absence and Melatonin Administration Restored Left Ventricular Lumen and Inhibited Thickening of Its Wall during Aging

Magnetic resonance imaging of the heart of the young (Y), early-aged (EA), and old-aged (OA) WT (Figure 1A–E) and NLRP3^−/−^ mice (Figure 1F–J), as well as the beneficial effect of melatonin supplementation was illustrated in Figure 1. With aging, there were no changes in the length of the heart of the EA WT and NLRP3^−/−^ mice; while the cardiac length displayed a significant decline in the OA WT mice. However, melatonin therapy significantly increased the cardiac length in the EA NLRP3^−/−^ animals, where its effect was more detectable than in WT mice (Table 1, Figure 1K). Moreover, aging induced a significant reduction in the length of the left ventricular lumen in the WT and NLRP3^−/−^ mice, and this decline was less detectable in the NLRP3^−/−^ than in the WT mice. Melatonin administration, however, recovered the luminal length of the left ventricle in the EA and OA WT and NLRP3^−/−^ mice (Table 1, Figure 1L). Aging was also accompanied with an increase in the thickness of the left ventricular wall in the WT and NLRP3^−/−^ mice, and this increase was less remarkable in the NLRP3^−/−^ than in the WT mice. Interestingly, this increase in the left ventricular wall thickness was countered in the OA animals by melatonin therapy, which revealed a more detectable beneficial effect on NLRP3^−/−^ mice (Table 1, Figure 1M).

### 3.2. NLRP3 Deficiency and Melatonin Therapy Enhanced Cardiac Anthropometric Parameters during Aging

Aging displayed a significant increase of body weight and heart weight in the WT and NLRP3^−/−^ mice, an effect enhanced by melatonin administration. While this induction in the body weight revealed no changes between WT and NLRP3^−/−^ mice, the increase of heart weight was higher in the NLRP3^−/−^ mice than WT ones with aging. Furthermore, melatonin therapy showed more considerable effects in the NLRP3^−/−^ mice than in WT animals (Table 1, Figure 1N,O). The ratio of the heart weight to the body weight, however, reported an age-mediated decline, which was higher in the OA WT mice than in the NLRP3^−/−^ ones. This reduction was significantly countered by melatonin supplementation (Table 1, Figure 1P).

### 3.3. NLRP3 Deletion and Melatonin Supplementation Reduced Age-Related Histological and Morphometrical Alterations of the Cardiac Myocytes, and Minimized Hypertrophy-Associated Genes as Well as Inflammatory Cytokines Genes

Histological analysis of the LV of the Y WT mice showed the normal architecture of the cardiac muscles, which consisted of cardiomyocytes in different orientations: longitudinal, transverse, and oblique. These cardiac fibers were separated from each other by narrow interstitial tissues, composed of blood capillaries, less collagenous tissue, and fibroblasts. The nucleus was centrally located (Figure 2A,B).

Cardiomyocytes of the EA WT mice revealed an initial degree of necrosis associated with lymphocytic infiltrates and widening of the interstitial spaces (Figure 2C,D). These alterations were progressive in the heart of OA animals, where the myocardium demonstrated severe degrees of necrotic damage, associated with lymphocytic infiltrations and excessive collagen deposition, an indicator of fibrosis. In addition, disorganization of the cardiac fibers and large interstitial spaces were illustrated (Figure 2G,H). Melatonin administration, however, elucidated a protective effect on the cardiomyocytes of both EA (Figure 2E,F) and OA (Figure 2I,J) WT mice. The fibers conserved their normal architecture with tight interstitium, less infiltrations of collagen, and absence of necrotic muscle fibers.

Histological examination of the cardiac muscles of the Y NLRP3^−/−^ mice (Figure 2K,L) showed the normal organization of the cardiac myocytes, which demonstrated no changes in the EA animals, except of wide interstitial tissues (Figure 2M,N), while less necrotic changes as well as collagen deposition were observed in OA ones (Figure 2Q,R), compared with those of WT mice. Interestingly, melatonin therapy induced more preservative effect on the cardiomyocytes of EA (Figure 2O,P) and OA (Figure 2S,T) NLRP3^−/−^ mice than WT mice, keeping the normal architecture of muscle fibers with narrow interstitial spaces.

Aging also induced a significant loss of cardiac muscle fibers, associated with hypertrophy of individual cardiomyocyte (increased cross-sectional area “CSA” of individual cardiomyocyte). This decline in cardiac fiber number and increase in cardiomyocyte CSA were more remarkable in WT mice than NLRP3^−/−^ ones. Melatonin therapy restored the number of cardiomyocytes and minimized muscle fiber hypertrophy in EA and OA WT and NLRP3^−/−^ mice (Figure 3A,B). Aging was associated with increased β-MHC expression in WT and NLRP3^−/−^ mice; however, the expression was more detectable in WT animals. Melatonin supplementation reduced β-MHC expression in both mice strain, with more considerable effect on NLRP3^−/−^ mice (Figure 3C).

Morphometrical analysis of the fibrotic area percentage revealed age-mediated induction of the cardiac fibrosis (Figure 3D), associated with increased expressions of IL-1α, IL-6 and TNFα in the old-aged animals, while non-significant increase was found in the early-aged mice (Figure 3E–G). These inductions of fibrosis and inflammatory cytokines genes were less remarkable in the NLRP3^−/−^ mice. Melatonin supplementation, however, reduced cardiac fibrosis and significantly decreased the expression of IL-1α and IL-6 in WT and NLRP3^−/−^ mice, with non-significant decline of TNFα.

### 3.4. NLRP3 Ablation and Melatonin Administration Conserved Cardiac Muscle Ultrastructure during Aging

Electron microscopy analysis of the LV of the Y WT mice clarified the normal ultrastructure of the cardiac muscle fibers, which are composed of centrally located nuclei and are formed as well-organized longitudinally arranged myofibrils that illustrated cross and longitudinal striations, with presence of the sarcoplasmic reticulum in between. Each myofibril constitutes of thread-like myofilaments: actin and myosin. The sarcomeres are well aligned between each two successive Z-lines. Cardiomyocytes branched repeatedly and attached strongly at the intercalated disc (Figure 4A). The mitochondria were intact and compacted with clearly organized cristae and were gathered in different orientations: clusters in between cardiac myofibrils as intermyofibrillar, around the nucleus, and beneath the sarcolemma as subsarcolemmal (Figure 4B). Cardiac fibers had better structure in Y NLRP3^−/−^ mice than WT ones. The myofibrils depicted an organized sarcomere and highly compacted mitochondria with densely packed, well-arranged cristae and narrow interstitial spaces with a normally oriented sarcoplasmic reticulum (Figure 4C,D).

At the early stage of aging, cardiac muscle fibers of WT mice showed disorganized myofibrils and sarcoplasmic reticulum. Some mitochondria displayed vacuolation and cristae damage. Appearance of small-sized multivesicular bodies, termed autophagosomes, was also detected (Figure 5A,B). Melatonin supplementation, however, preserved the cardiac myocytes, maintained the normal orientation of the myofibrils and the intact contents of mitochondria, and reduced the residual bodies (Figure 5C,D). In contrast, cardiac muscle fibers of the EA NLRP3^−/−^ did not show age-related alterations in the myofibrils architecture and mitochondrial composition, except presence of individual disorganized myofibrils with indistinct striations (Figure 5E,F), which was improved with melatonin therapy that revealed a better preservative effect in EA NLRP3^−/−^ mice than WT ones (Figure 5G,H).

The cardiac muscle fibers of the OA WT mice demonstrated severe damage of myofibrils, with widening of interstitial spaces and disruption of the sarcoplasmic reticulum. Some mitochondria revealed a normal structure, while others were hypertrophied and demonstrated different stages of cristae damage and presence of inclusion bodies on their matrix. Splitting of the nucleus into two or three parts was mostly detected, with formation of autophagosomes (Figure 6A–C). Meanwhile, melatonin supplementation conserved the cardiac muscle constitutions and protected nuclear and mitochondrial contents, except of individual mitochondria that displayed destructed damage. Small-sized residual bodies were also found (Figure 6D–F).

The severe age-associated changes detected in cardiac myocytes were less detectable in OA NLRP3^−/−^ mice than WT ones. The cardiac muscles of the OA NLRP3^−/−^ were of less prevalent muscular damage, with lipid droplets. Some mitochondria were normal and intact, while others were characterized by their widely separated and disorganized cristae (Figure 6G,H). Melatonin administration induced a beneficial effect, where it maintained normal muscular structure and mitochondrial architecture during aging. Most of the myofibrils and mitochondria were intact, while individual fibers revealed residual bodies and lipid infiltrations (Figure 6I,J).

### 3.5. NLRP3 Absence and Melatonin Treatment Diminished Cardiac Apoptosis during Aging

Hoechst fluorescent analysis of the cardiac muscle fibers nuclei showed the normal nuclear appearance in Y WT and NLRP3^−/−^ mice (Figure 7A,B). With aging, cardiac muscle fibers of EA WT and NLRP3^−/−^ animals illustrated signs of apoptosis, where apoptotic cells illustrated cell shrinkage, chromatin condensation, and nuclear fragmentation (Figure 7C,D). These age-related apoptotic changes were more pronounced in the OA groups (Figure 7G,H) and were more considerable in cardiac myocytes of WT mice than NLRP3^−/−^ one (Figure 7K). Interestingly, this age-associated increase of the nuclear apoptosis was countered with melatonin therapy in EA (Figure 7E,F) and OA (Figure 7I,J) WT and NLRP3^−/−^ mice, respectively (Figure 7K).

## 4. Discussion

This study describes for the first time the contribution of NLRP3 inflammasome to heart deterioration during aging. The NLRP3 inflammasome plays an essential role in the pathogenesis of various cardiovascular diseases such as hypertension, atherosclerosis, and myocardial infarction [14,47]. To better understand the role of NLRP3 in cardiac aging and age-related cardiac sarcopenia, we examined the left ventricle of different aged WT and NLRP3^−/−^ mice. As observed here, the deletion of the NLRP3 inflammasome resulted in a better cardiac architecture with no age-associated changes in the EA mice and with less necrotic and fibrotic changes in the OA animals when compared by WT mice. Recent study performed on NLRP3-knockout mice revealed a significant increase of heart weight/body weight ratio in the old WT mice, while non-significant increases were detected in old NLRP3^−/−^ ones [48]; however, this ratio of the heart weight to the body weight as observed here reported an age-mediated reduction, confirming the previous finding reported elsewhere [49], and this decline was higher in the old-aged WT mice than in the NLRP3^−/−^ ones.

The reduction of heart weight to body weight during aging was associated with decline in cardiac fiber numbers, increased left ventricular wall thickness, and an enhanced compensated cardiomyocyte hypertrophy of the remaining fibers with increased β-MHC mRNA expression. These alterations were less detectable in NLRP3^−/−^ mice than WT ones. Recently, increased cardiomyocyte transverse cross-sectional area was reported in aged WT mice unlike NLRP3^−/−^ ones [48]. Our previous study in gastrocnemius muscle confirmed these results, where lack of NLRP3 inflammasome showed lower muscular decline and reduced collagen fibers in aged NLRP3^−/−^ mice when compared with aged WT ones [31]. The increased expression level of β-MHC during cardiac aging was previously reported [50]. Another study suggested that β-MHC expression during aging is a marker of fibrosis rather than of cellular hypertrophy [51].

Recently, a study revealed increased mass and collagen level of the left ventricle, as well as the thickness of the septal wall in aged mice, associated with increased expressions of IL-1α, IL-6 and TNFα, suggesting that cardiac structural and functional changes with age are closely graded with frailty and inflammation markers [52]. NLRP3 inflammasome-related inflammaging has been activated with age in most body tissues and organs including heart [15,16]. Pro-inflammatory cytokines induced through age-dependent inflammasome activation promote muscle tissue wasting and atrophy, whilst lack of NLRP3 protects against these inflammatory proceedings [53]. The induction of fibrosis and IL-1α and IL-6 inflammatory cytokines genes observed here were less remarkable in the NLRP3^−/−^ mice. Recently, interstitial and perivascular cardiac fibrosis was described in aged WT mice, with non-significant changes in aged NLRP3^−/−^ ones, where increased IL-6 serum and protein levels in the cardiac tissues of old WT and NLRP3^−/−^ mice were observed [48], and thus, the reduction of collagenous tissue infiltrations in the aged myocardium of NLRP3^−/−^ mice assured the attenuation of fibrosis upon NLRP3 depletion [52].

Mitochondria play critical roles in cellular life and death, as it is important for the cellular homeostatic maintenance, and therefore mitochondrial dysfunction with aging has been implicated in the deterioration in structure and function of skeletal and cardiac muscles [54]. The correlation between mitochondrial function and cardiovascular health has been recently investigated, where lower mitochondrial oxidative capacity in aged individuals was associated with a positive previous history of cardiovascular disease events [55]. Furthermore, poorer mitochondrial function was recently proposed as a potential contributor of increased perceived fatigability [56]. As shown by electron microscopy, the current study demonstrated that cardiac muscle aging is associated by a diversity of ultrastructural alterations, including mitochondrial swellings, cristae destruction and matrix vacuolization, and increased lipid accumulations as well as decline in mitochondrial number previously detected [37]. These observations of the mitochondrial ultrastructural alterations have been proposed as an indicator of cellular senescence and age-dependent loss of mitochondrial functions [29,57] and were less considerable in aged cardiomyocytes of NLRP3^−/−^ mice than WT [37,48], confirming our previous findings in the skeletal muscle, where the lack of NLRP3 inflammasome reduced mitochondrial impairment during aging [31] and also supporting the role of NLRP3 inflammasome inhibition in prevention of cardiac aging [58]. Our recent study showed alteration of mitochondrial dynamics in cardiac muscles during aging, where the level of proteins involved in mitochondrial dynamics such as Mfn2, Opa1 and Drp1 revealed a significant decline with age in WT mice; meanwhile, this decline was absent in NLRP3^−/−^ mice [37].

Aging of cardiomyocytes was associated with formation of autophagosomes, which were more pronounced in aged cardiomyocytes of WT mice than NLRP3^−/−^ mice. Many studies have suggested the involvement of autophagy in the regulation of lifespan and aging [59]. It plays an essential role in mitigation of age-associated cardiac changes [60]. The age-dependent decline of autophagy in the heart [61] enhances impairments in cellular housekeeping functions that induce NF-κB signaling, which either directly or through inflammasomes stimulates age-related pro-inflammatory events [62]. Moreover, aging diminishes the autophagic/mitophagic capacity and thus leads to an accumulation of reactive oxygen species, which triggers the activation of NLRP3 inflammasome and induces inflammation in various tissues [52,62], while absence of NLRP3 improves mitochondrial dysfunction. Therefore, our results revealed reduction of autophagosome numbers and size in aged cardiac muscles of NLRP3^−/−^ mice, confirming the beneficial effect of NLRP3 inhibition in improvement of autophagy quality during cardiac aging, where NLRP3 inflammasome deletion protected mice against age-dependent induction of insulin sensitivity, avoiding a wide variety of age-associated cardiac changes, preserving cardiac functions of aged animals, and consequently, prolonging lifespan [58]. Our previous study reported a significant decline of LC3II/LC3I ratio, a hallmark of autophagy, in cardiac muscles of early-aged and old-aged WT mice; meanwhile, loss of NLRP3 had few impacts on age-dependent cardiac autophagic changes [37]. A recent study confirmed the beneficial effect of NLRP3 inhibition for enhancement of the autophagy in aged mice. Moreover, this latter study proposed the impact of NLRP3 suppression on improving human health and age-dependent metabolic syndrome [48].

Aging of cardiac muscle fibers was associated with increased nuclear apoptosis, which was more pronounced in cardiac myocytes of WT mice than NLRP3^−/−^ ones. Apoptosis displays a key role in the muscular loss, where it has been defined in muscular denervation [63] as well as in chronic heart failure [64]. It is an active process causing designed cellular death and is associated with removal of apoptotic bodies without inflammation through phagocytosis of these bodies by surrounding cells or macrophages [65]. An increase in apoptosis and necrosis was previously observed in the myocardium of old animals [66]. Our recent study detected less cardiac apoptosis during aging in the absence of NLRP3, where aging induced Bax/Bcl2 ratio in the cardiomyocytes of WT mice, while this ratio remained without changes in cardiac muscle of aged NLRP3^−/−^ mice [37].

Besides its chronobiotic properties due to the low-level circadian release of melatonin by the pineal gland, melatonin is also produced in most organs and tissues of the body [21]. The so-called extrapineal melatonin exerts profound antioxidant and anti-inflammatory actions due to high levels that reach the cells [21]. Recently, it has been shown that mouse heart produces high amounts of melatonin that contributes to cardiac protection [22,67]. Cardiac melatonin, however, decreases with age [68], thus reducing the cardioprotection that it may promote. Here, melatonin supplementation to mice prevented the progress of age-related cardiac sarcopenia, where melatonin conserved the normal architecture of cardiomyocytes with narrow interstitium, less fibrosis, and absence of necrotic fibers. Moreover, melatonin treatment recovered the thickness of the ventricular wall, improved the ratio of heart weight/body weight through restoring cardiomyocytes number, and minimized muscle fiber hypertrophy. These results were in accordance with a decrease in β-MHC expression, which, interestingly, was more pronounced in NLRP3^−/−^ compared to WT mice. This higher effectiveness of melatonin in some parameters of mutant mice could be due to an adjuvant effect of melatonin and lack of NLRP3 inflammasome. The adjuvant effect of melatonin has been used in several therapies against different pathologies including atherosclerotic carotid arterial stenosis, cancer, COVID-19, and diabetic neuropathy [69,70,71,72]. Recently, activation of mitochondrial calcium uptake 1 (MICU1) induced by melatonin has been suggested as a mechanism underlying the protective effect of melatonin against cardiac hypertrophy [73].

Melatonin showed an anti-apoptotic action in EA and OA animals. Furthermore, melatonin supplementation significantly decreased the expression of IL-1α and IL-6 in WT and NLRP3^−/−^ mice, confirming the anti-oxidative, anti-apoptotic, and anti-inflammatory effects of melatonin [21,74]. Melatonin administration was reported to inhibit cardiomyocytes apoptosis and improve organization of the actin filament as well as maintain calcium hemostasis as protective mechanisms against myocardial reperfusion injury [75]. The beneficial effects of melatonin treatment against age-related cardiac apoptosis and autophagic changes was more considerable in NLRP3^−/−^ mice than WT ones [37]. Our recent study showed that lack of NLRP3 inflammasome improved the arrangement of murine skeletal muscle fibers and reduced the deposition of collagenous tissues compared with WT muscle during aging [76].

Mitochondria are the key intracellular target of melatonin, which reduces free radical’s formation and boosts the ATP production in both normal and pathological conditions [35,77,78,79]. Melatonin therapy protected the cardiac muscle fibers against the age-dependent mitochondrial damage. It maintained the normal ultrastructure of cardiac myocytes, preserved the mitochondrial contents, reduced residual and multivesicular bodies, and also kept the integrity of the IMF mitochondrial number with aging. These preservative effects of melatonin were more detectable in NLRP3^−/−^ mice than in WT ones [37], confirming our previous findings, where melatonin protected muscles from age-related sarcopenia-dependent mitochondrial damages, and NLRP3 inflammasome deletion reduced these alterations and induced the protective effects of melatonin [25,26]. Recently, the beneficial effect of melatonin on countering the decline of Mfn2, Opa1, and Drp1 protein levels was reported in aged cardiac muscle of WT mice [37]. All of these findings support essential impacts of melatonin on preventing mitochondrial dysfunction, reducing oxidative stress, and minimizing sarcopenic alterations in patients [80].

## 5. Conclusions

In conclusion, our study clarifies the early morphological markers of cardiac aging and also supports the role of the NLRP3 inflammasome in the development of age-dependent cardiac sarcopenia, in addition to the protective effect of melatonin supplementation in prevention of cardiac aging (Figure 8). The findings of this study may be summarized as follows: (1) less induction of the ventricular wall thickness and lower decline of ratio of heart weight to body weight in in aged NLRP3^−/−^ mice than in aged WT mice, clarifying the contribution of the NLRP3 inflammasome to sarcopenia; (2) fewer collagenous fibers deposition and less IL-1α and IL-6 expressions in the aged myocardium of NLRP3^−/−^ mice than in aged WT mice, assuring the role of NLRP3 in fibrosis induction; (3) reduced mitochondrial damage and multivesicular bodies’ formation in cardiac muscle fibers of aged NLRP3^−/−^ mice compared with aged WT mice, confirming the role of NLRP3 ablation in amelioration of mitochondrial dysfunction; (4) fewer autophagosome accumulation and size in aged cardiomyocytes of NLRP3^−/−^ mice than in aged WT mice, revealing the role of NLRP3 inhibition in improvement of autophagy quality during cardiac aging; (5) the prophylactic effects of melatonin against age-related muscular necrosis and fibrosis, thickening of the ventricular wall, decline of myocytes number, induction of myocytes apoptosis, and increased expressions of β-MHC, IL-1α and IL-6; (6) the protective effect of melatonin against age-associated autophagosomes formation and accumulation, affirming its role in induction of autophagic processes and prohibition of muscular myopathy and/or denervation; (7) higher beneficial effect of melatonin against age-accompanied sarcopenic changes in cardiac muscles of NLRP3^−/−^ mice than in WT mice, supporting the dependence of melatonin effect at least against the activation of the NLRP3 inflammasome. Further investigations should be carried out to elucidate the exact mechanism by which melatonin exerts more effective actions in certain parameters of the NLRP3^−/−^ mice.

## 6. Limitation

The heart weight was normalized here to the body weight; however, the body weight changes with age, where some studies reported increase of HW/BW during aging, while others revealed non-significant decline with age. Although normalization of the heart weight to the tibial length is very important to overcome this conflict, unfortunately, we could not get these data because of lack of the old-aged group and treated ones.

## Figures and Tables

**Figure 1 antioxidants-10-01269-f001:**
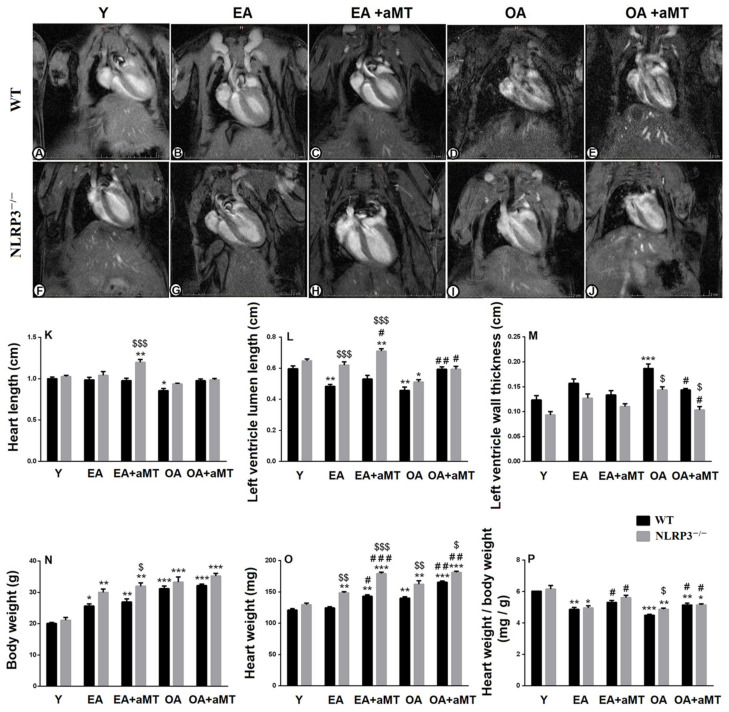
Impact of NLRP3 deficiency and melatonin therapy on cardiac magnetic resonance imaging and anthropometric parameters during aging. (**A**–**E**) Magnetic resonance imaging of the heart of young (Y), early-aged (EA), early-aged with melatonin (EA + aMT), old-aged (OA), and old-aged with melatonin (OA + aMT) WT mice. (**F**–**J**) Magnetic resonance imaging of the heart of Y, EA, EA + aMT, OA, and OA + aMT NLRP3^−/−^ mice. (**K**) Analysis of the heart length (cm). (**L**) Analysis of the luminal length of the left ventricle (cm). (**M**) Analysis of the thickness of the left ventricular wall (cm). (**N**) Analysis of the body weight (g). (**O**) Analysis of the heart weight (mg). (**P**) Analysis of the ratio of the heart weight to the body weight (mg/g). * *p* < 0.05, ** *p* < 0.01 and *** *p* < 0.001 vs. Y; # *p* < 0.05, ## *p* < 0.01 and ### *p* < 0.001 vs. aged group without melatonin; $ *p* < 0.05, $$ *p* < 0.01 and $$$ *p* < 0.001 vs. WT mice.

**Figure 2 antioxidants-10-01269-f002:**
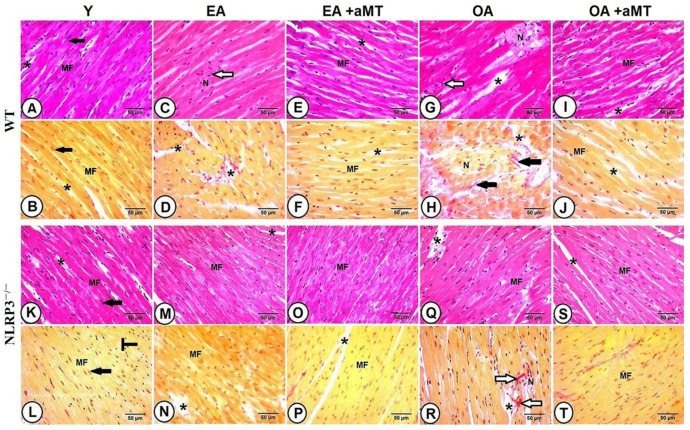
Effect of NLRP3 deletion and melatonin supplementation on age-associated histological changes of cardiac muscle fibers. (**A**,**B**) Left ventricle (LV) of the Y WT mice showing the normal cardiac muscle fibers (MF) architecture, with central nucleus (black arrows), and narrow interstitial spaces (black asterisks). (**C**,**D**) LV of the EA WT mice revealing an initial necrotic degree (N), with lymphocytic infilterates (white arrow) and wide interstitial spaces (black asterisks). (**E**,**F**) The conservative effect of melatonin on maintaining normal muscle fibers (MF) of EA mice, with narrow interstitial spaces (black asterisks). (**G**,**H**) LV of the OA WT mice demonstrating severe necrotic changes (N), with lymphocytic infiltrations (white arrow), excessive collagen deposition (black arrows), and wide interstitial spaces (black asterisks). (**I**,**J**) The protective effect of melatonin on improving cardiac muscle fibers (MF) and minimizing interstitial tissues (black asterisks) in OA animals. (**K**,**L**) LV of the Y NLRP3^−/−^ mice showing the normal cardiac muscle fibers organization (MF), with centrally located nuscle (black arrows) and interstitial spaces (black asterisks). (**M**,**N**) LV of the EA NLRP3^−/−^ mice revealing wide interstitial spaces (black asterisks) without necrosis. (**O**,**P**) The preservative effect of melatonin on keeping normal structure of cardiac myocytes (MF) in EA mice. (**Q**,**R**) LV of the OA NLRP3^−/−^ mice demonstrating less necrotic changes (N), interstitail spaces (black asterisks), and collagen deposition (white arrows). (**S**,**T**) The beneficial effect of melatonin on improving cardiac architecture, with narrow interstitial spaces (black asterisk). Bar = 50 μm. (**A**,**C**,**E**,**G**,**I**,**K**,**M**,**O**,**Q**,**S**) stained with H&E stain, while (**B**,**D**,**F**,**H**,**J**,**L**,**N**,**P**,**R**,**T**) stained with Van Gieson stain.

**Figure 3 antioxidants-10-01269-f003:**
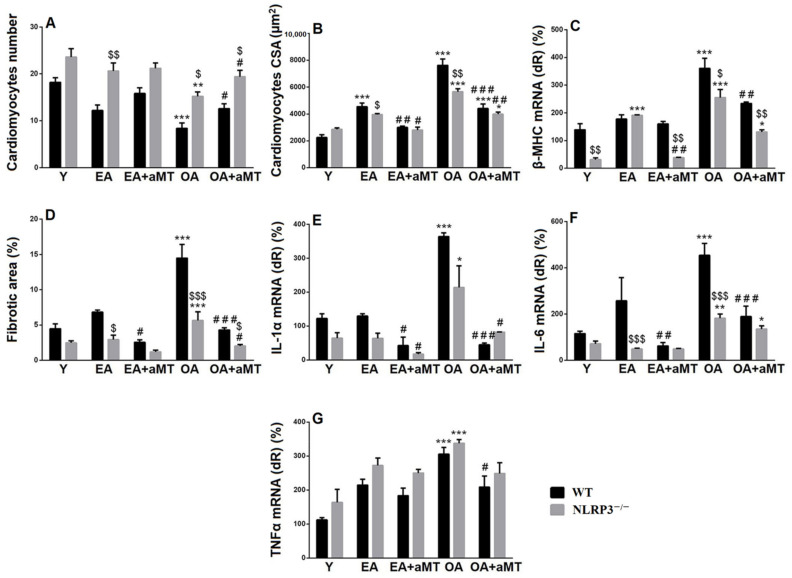
Effect of NLRP3 deletion and melatonin supplementation on age-associated morphometrical changes of cardiac muscle fibers, as well as hypertrophy-associated and inflammatory cytokines genes. (**A**,**B**) Age-associated morphometrical changes in cardiac muscle fibers number (per 100 μm^2^) and cross-sectional area (CSA). (**C**) mRNA expression level of β-MHC. (**D**) Morphometrical analysis of cardiac fibrosis during aging. (**E**) mRNA expression level of IL-1α. (**F**) mRNA expression level of IL-6. (**G**) mRNA expression level of TNFα * *p* < 0.05, ** *p* < 0.01 and *** *p* < 0.001 vs. Y; # *p* < 0.05, ## *p* < 0.01 and ### *p* < 0.001 vs. aged group without melatonin; $ *p* < 0.05, $$ *p* < 0.01 and $$$ *p* < 0.001 vs. WT mice.

**Figure 4 antioxidants-10-01269-f004:**
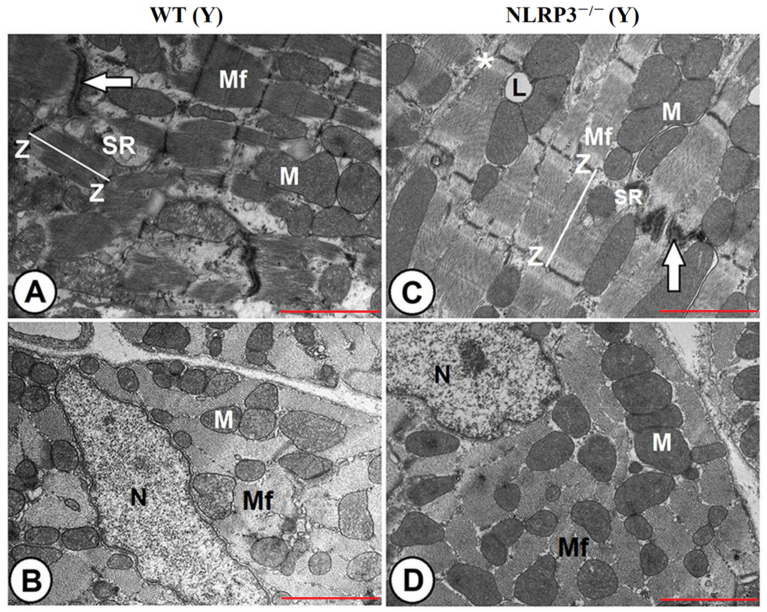
Ultrastructural architecture of cardiac muscle fibers in young WT and NLRP3^−/−^ mice. (**A**,**B**) Transmission electron micrographs of the LV of the Y WT mice clarifying well-organized longitudinally arranged myofibrils (Mf), with cross and longitudinal striations, centrally located nucleus (N), and intermyofibrillar sarcoplasmic reticulum (SR). Sarcomeres (line) are well aligned between each two successive Z-lines (Z), and cardiomyocytes branched repeatedly and attached strongly at the intercalated disc (arrow). The mitochondria were intact and compacted with clearly organized cristae, (M). (**C**,**D**) Transmission electron micrographs of the LV of the Y NLRP^−/−^ mice showing cardiac myofibrils (Mf) of better structure with organized sarcomere between Z-lines (Z), highly compacted mitochondria (M), narrow interstitial spaces (asterisk), intermyofibrillar distribution of sarcoplasmic reticulum (SR), and lipid droplets (L). Nucleus (N) was centrally located. Bar “red line” = 2 μm.

**Figure 5 antioxidants-10-01269-f005:**
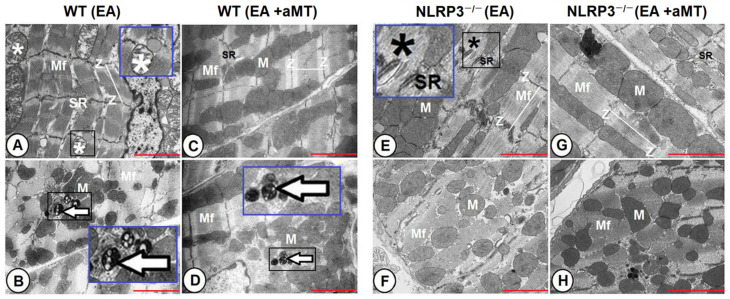
Impact of NLRP3 absence and melatonin treatment on the ultrastructure of early-aged cardiac myocytes. (**A**,**B**) Transmission electron micrographs of the LV of the EA WT mice showing disorganized myofibrils (Mf) and sarcoplasmic reticulum (SR). Some mitochondria displayed vacuolation and cristae damage (white asterisk), with appearance of small-sized multivesicular bodies (arrow). Higher magnifications of the black boxed areas in the blue insets. (**C**,**D**) Transmission electron micrographs of the LV of the EA WT mice after melatonin supplementation. The cardiac myofibrils (Mf) maintained the normal orientation, with intact mitochondrial contents (M), organized sarcoplasmic reticulum (SR), and reduced residual bodies (arrow). Higher magnification of the black boxed area in the blue inset. (**E**,**F**) Transmission electron micrographs of the LV of the EA NLRP^−/−^ mice revealing absence of age-related alterations in the myofibrils architecture (Mf), sarcoplasmic reticulum (SR), and mitochondrial composition (M), except presence of individual disorganized myofibrils with indistinct striations (black asterisk). Higher magnification of the black boxed area in the blue inset. (**G**,**H**) Transmission electron micrographs of the LV of the EA NLRP^−/−^ mice with melatonin therapy demonstrating improvement of myofibrils (Mf) and their sarcoplasmic reticulum (SR) and mitochondrial contents (M). Note sarcomeres arrangement between each two successive Z-lines (Z). Bar “red line” = 2 μm.

**Figure 6 antioxidants-10-01269-f006:**
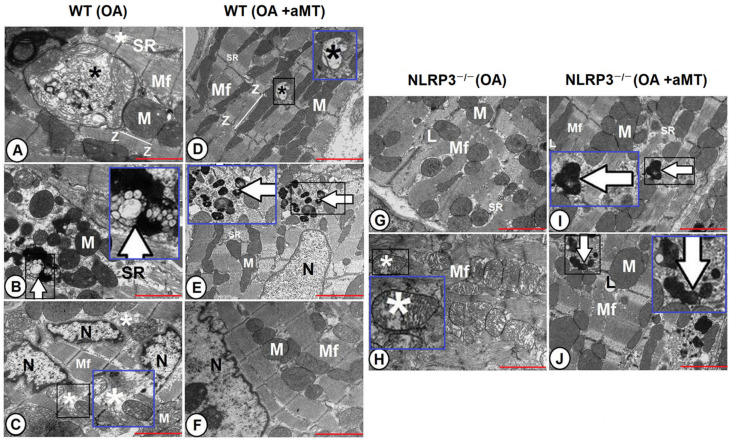
Effect of NLRP3 ablation and melatonin administration on age-associated ultrastructural alterations of cardiac muscle fibers. (**A**–**C**) Transmission electron micrographs of the LV of the OA WT mice illustrating widening of interstitial spaces (white asterisk) and disruption of the sarcoplasmic reticulum (SR) between myofibrils (Mf). Some mitochondria revealed a normal structure (M), while others were hypertrophied and demonstrated different stages of cristae damage with presence of inclusions bodies on their matrix (black asterisks). Splitting of the nucleus (N) was mostly clarified, with formation of autophagosomes (arrow). Higher magnifications of the black boxed areas in the blue insets. (**D**–**F**) Transmission electron micrographs of the LV of the OA WT mice after melatonin supplementation showing conservation of cardiac myofibrils (Mf), nuclear structure (N), sarcoplasmic reticulum (SR), and mitochondrial contents (M), except of individual mitochondria displayed destructed damage (black asterisk), in addition to presence of small-sized residual bodies (arrow). Higher magnifications of the black boxed areas in the blue insets. (**G**,**H**) Transmission electron micrographs of the LV of the OA NLRP3^−/−^ mice illustrating less prevalent damage of cardiac myofibrils (Mf), with well-organized sarcoplasmic reticulum (SR) and lipid droplet (L) infiltrations. Some mitochondria were normal, and intact, while others were characterized by their widely separated and organized cristae (asterisk). Higher magnification of the black boxed area in the blue inset. (**I**,**J**) Transmission electron micrographs of the LV of the OA NLRP3^−/−^ mice after treatment with melatonin depicting the maintenance of normal myofibrils (Mf) structure, sarcoplasmic reticulum (SR) organization, and mitochondrial architecture (M) during aging. Individual fibers containing residual bodies (arrows) and lipid infiltrations (L) were observed. Higher magnifications of the black boxed areas in the blue insets. Note organization of sarcomeres between each two successive Z-lines (Z). Bar “red line” = 2 μm.

**Figure 7 antioxidants-10-01269-f007:**
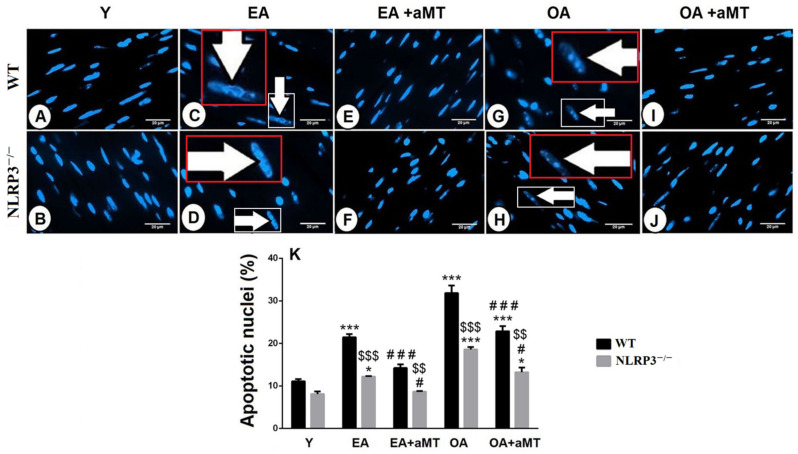
Impact of NLRP3 deletion and melatonin therapy on age-dependent cardiac apoptosis. (**A**,**B**) Hoechst fluorescent analysis of the nuclear apoptosis in the LV of the Y WT and NLRP3^−/−^ mice, respectively showing normal nuclear appearance. (**C**,**D**) Analysis of the nuclear apoptosis in the LV of the EA WT and NLRP3^−/−^ mice, respectively revealing that aging induced nuclear apoptosis and fragmentation (arrows). Higher magnifications of the white boxed areas in the red insets. (**E**,**F**) Analysis of the nuclear apoptosis in the LV of the EA WT and NLRP3^−/−^ mice, respectively after melatonin therapy clarifying the protective effect of melatonin on reducing cardiac apoptosis. (**G**,**H**) Analysis of the nuclear apoptosis in the LV of the OA WT and NLRP3^−/−^ mice, respectively exhibiting more detectable nuclear apoptosis. Higher magnifications of the white boxed areas in the red insets. (**I**,**J**) Analysis of the nuclear apoptosis in the LV of the OA WT and NLRP3^−/−^ mice, respectively after melatonin treatment confirming the beneficial effect of melatonin against age-related cardiac apoptosis. **(K)** Morphometriacal analysis of apoptotic nuclei of cardiomyocytes during aging revealing that cardiac apoptosis was more considerable in cardiac myocytes of WT mice than in those of NLRP3^−/−^ one. Bar “white line” = 20 μm. * *p* < 0.05 and *** *p* < 0.001 vs. Y; # *p* < 0.05 and ### *p* < 0.001 vs. aged group without melatonin; $$ *p* < 0.01 and $$$ *p* < 0.001 vs. WT mice.

**Figure 8 antioxidants-10-01269-f008:**
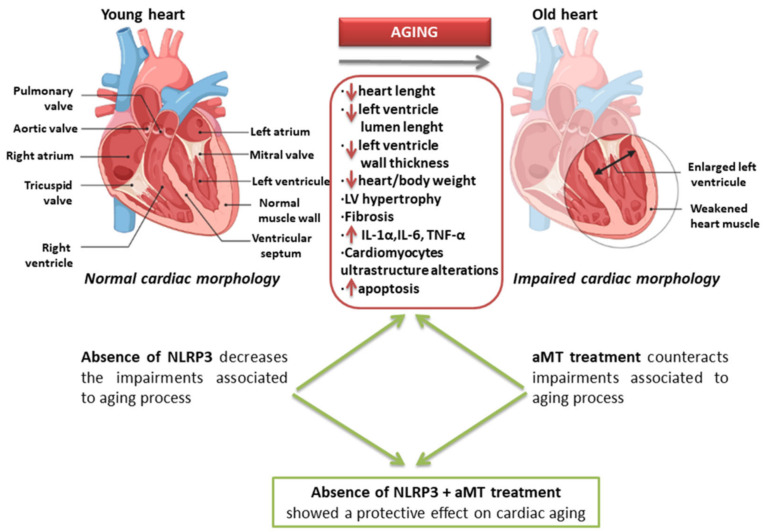
Schematic graph summarizing the age-associated alterations of the cardiac muscle fibers and the beneficial effects of NLRP3 inflammasome deletion and melatonin (aMT) supplementation.

**Table 1 antioxidants-10-01269-t001:** Morphometrical analyses of MRI and anthropometric parameters during aging.

Parameter	Wild-Type Mice					NLRP3-Knockout Mice				
	Y	EA	EA + aMT	OA	OA + aMT	Y	EA	EA + aMT	OA	OA + aMT
**Heart length (cm)**	1.00 ± 0.02	0.99 ± 0.03	0.98 ± 0.03	0.86 ± 0.02	0.98 ± 0.02	1.03 ± 0.01	1.04 ± 0.05	1.2 ± 0.03	0.94 ± 0.01	0.99 ± 0.01
**Left ventricle lumen length (cm)**	0.6 ± 0.02	0.48 ± 0.01	0.53 ± 0.02	0.46 ± 0.02	0.59 ± 0.01	0.65 ± 0.01	0.62 ± 0.02	0.71 ± 0.02	0.51 ± 0.01	0.59 ± 0.02
**Left ventricle wall thickness (cm)**	0.12 ± 0.01	0.16 ± 0.008	0.13 ± 0.006	0.19 ± 0.008	0.14 ± 0.003	0.09 ± 0.006	0.13 ± 0.008	0.11 ± 0.005	0.14 ± 0.007	0.10 ± 0.007
**Body weight (g)**	20.07 ± 0.35	25.63 ± 0.66	26.93 ± 0.97	31.17 ± 0.84	32.17 ± 0.47	21.10 ± 0.92	30 ± 1.02	31.93 ± 1.14	33.27 ± 1.68	35.23 ± 0.87
**Heart weight (mg)**	120.8 ± 2.3	124.4 ± 1.7	142.9 ± 2.6	140.1 ± 2.2	165.3 ± 2.1	129.5 ± 2.5	148.4 ± 1.9	179.0 ± 2.3	161.9 ± 5.9	181.1 ± 2
**Heart weight/body weight (mg/g)**	6.02 ± 0.01	4.86 ± 0.13	5.32 ± 0.12	4.5 ± 0.05	5.14 ± 0.12	6.15 ± 0.23	4.96 ± 0.14	5.62 ± 0.15	4.88 ± 0.07	5.15 ± 0.08

Y, young, EA, early-aged; EA + aMT, early-aged with melatonin; OA, old-aged; OA + aMT, old-aged with melatonin.

## Data Availability

The data presented in this study are available on request from the corresponding author. The data are not publicly available due to they are not in any public repository.

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
