# Peer review of "The Impact of Melatonin Supplementation and NLRP3 Inflammasome Deletion on Age-Accompanied Cardiac Damage"

_antioxidants, 2021, doi:10.3390/antiox10081269_

Round 1
Reviewer 1 Report
The Authors in the present original article evaluated early cardiac morphological changes related to the physiopathological process of aging and described also the NLRP3 inflammasome involvment in age-related cardiac sarcopenia. Furthrmore, They described the protective effects of melatonin against age-dependent cardiac morphological and ultrastructural alterations.
Good article. I think that the present manuscript has an interesting and actual topic and it could be a starting point for a better understanding of the cardiac aging physiopathological process and, interestingly, it provided evidence that melatonin treatment is able to prevent heart morphological aging-related injury.
In my opinion, this manuscript has a clear message, the rationale for the choice of the experimental model as well as the technical approaches used are appropriate. The obtained results are fully described and discussed.
However, I have some mandatory comments:
- In all the text check abbreviations (such as at lines 78 and 96).
Title
- In my opinion a succinct title will be more attactive for the readers.
Introduction
- Lines 86-88: due to melatonin is a multitasking molecule, brief specify its multiple properties and not only the anti-inflammatory and anti-oxidative features;
- Specify in the mn aim the animal model.
Materials and Methods
- Line 128: add one or more references that justify the melatonin dose and time of treatment;
- Lines 178-187: specify if the % of apototic nuclei was evalueted in blind and by 2 observers (as for the morphometrical analyses).
Results
- Figures 1K-P: a table that summazies the data reported in figs 1k-p will be useful for the readers;
- Figures 4-6: zooms, near the main photomicrographs, that show in deep the aging-related ultrastructural alteration will help the readers. The scale bars are too small to be read;
- Figure 7: zooms, near the main photomicrographs, that show in deep the nuclear injury will help the readers (alterations identify with the arrows).
Discussion
- The animals used are female, sex differences may be observed with male mice?
- the Discussion paragraph is too long;
- a simple schematic graph that summarized the data obtain should be useful for the readers.
Author Response
Amendments to the questions rose by the reviewer #1 to the manuscript ID antioxidants-1313710
- The Authors in the present original article evaluated early cardiac morphological changes related to the physiopathological process of aging and described also the NLRP3 inflammasome involvement in age-related cardiac sarcopenia. Furthermore, they described the protective effects of melatonin against age-dependent cardiac morphological and ultrastructural alterations.
- Good article. I think that the present manuscript has an interesting and actual topic and it could be a starting point for a better understanding of the cardiac aging physiopathological process and, interestingly, it provided evidence that melatonin treatment is able to prevent heart morphological aging-related injury.
- In my opinion, this manuscript has a clear message, the rationale for the choice of the experimental model as well as the technical approaches used are appropriate. The obtained results are fully described and discussed.
- However, I have some mandatory comments:
- In all the text check abbreviations (such as at lines 78 and 96).
The abbreviations were revised.
Title
- In my opinion a succinct title will be more attractive for the readers.
The title was revised accordingly.
Introduction
- Lines 86-88: due to melatonin is a multitasking molecule, brief specify its multiple properties and not only the anti-inflammatory and anti-oxidative features.
This issue was revised and multiple properties of melatonin were included on page 2, paragraph 6.
- Specify in the mn aim the animal model.
This issue was revised and the animal model used in this study was included in the introduction page 3, end of first paragrap. Moreover, more information about mice strain used in this study was included in Materials and Methods section, first paragraph.
Materials and Methods
- Line 128: add one or more references that justify the melatonin dose and time of treatment.
This issue was revised and more data about melatonin dose was included on materials and methods section page 3, paragraph 3.
- Lines 178-187: specify if the % of apoptotic nuclei was evaluated in blind and by 2 observers (as for the morphometrical analyses).
This issue was revised and corrected accordingly, page 4, line 276.
Results
- Figures 1K-P: a table that summarizes the data reported in figs 1k-p will be useful for the readers.
The table was included in the results section as a Table 1 on page 6.
- Figures 4-6: zooms, near the main photomicrographs, that show in deep the aging-related ultrastructural alteration will help the readers. The scale bars are too small to be read.
The figures were revised accordingly and magnifications of the ultrastructural alterations were included in the figures within an inset. Furthermore, the color of scale bars was changed into red to be more visible, and also described in the figure legend.
- Figure 7: zooms, near the main photomicrographs, that show in deep the nuclear injury will help the readers (alterations identify with the arrows).
The figure was revised and magnifications of the ultrastructural alterations were included in the figures within an inset.
Discussion
- The animals used are female, sex differences may be observed with male mice?
We are grateful to the reviewer concerning this comment. In fact, we studied here only female animals, and thus we couldn’t assure if sex differences could be observed in cardiac muscle fibers during aging or not.
In this regard, previous study in human revealed sex-specific variation in age-associated remodeling, where the decline of ventricular myocytes number with age was reported through apoptosis in men but not in women (PMID: 27395082). However, recent study conducted on male WT mice detected age-associated alterations of cardiac myocytes similar to those described here in female WT mice. These alterations include increased cardiomyocyte cross-sectional area, interstitial and perivascular fibrosis, mitochondrial disarray, degeneration, fragmentation, reduction of mitochondrial area, and disorganization of mitochondrial cristae (PMID: 31625260).
- The Discussion paragraph is too long.
Although we revised the Discussion to attempt to reduce it, it is very difficult due to the number of age-associated alterations being discussed. Moreover, various studies regarding the impact of NLRP3 inflammasome and melatonin treatment on cardiac aging were addressed. Therefore, it was difficult to make more reduction of this section to assure readers could get from this version enough information about cardiac aging.
- A simple schematic graph that summarized the data obtain should be useful for the readers.
This point has been addressed and a schematic graph that summarized the results has been added in the conclusion section as a Figure 8 on page 16.
Reviewer 2 Report
This is an interesting study showing the involvement of the NLRP3 inflammasome in mediating age-relate changes in the cardiac muscle, through the study of the lack of NLRP3. The study has the potential to add useful information to the field if the authors are able to answer some of the below issues. Much is based on observational data and would benefit from more biochemical and functional studies as I highlight below.
Major points:
- Can HW/Tibia length also be reported? This is important as body weight changes with age. If tibia length not available, this should be mentioned as a limitation. This is also important since there appears to be conflicting reports in the literature with one report showing an increase (ref 34) and another showing a reduction (ref 35) in old WT mice.
- What about Il-1b and IL-18mRNA? Also it would be interesting to know what happens at the protein level with respect to pro-IL1b and Pro-IL18 and the secreted forms. I suggest Western blots to detect both species. Gasdermin D should also be measured.
- Results in aging mitochondria are really interesting. It would be interesting to probe the biochemistry and functional changes behind the observed structural changes. How is OxPhos affected? Suggest performing Seahorse or Oroboros-type experiments to address this. Expression of genes like Opa1 would add to the story. Are ROS levels affected in the aging heart and does NLRP3-/- or melatonin restore this?
- Autophagy results also interesting but again more is needed to confirm this. Some genes worth looking at include: Beclin1, autophagy related gene 5 (Atg5), and microtubule-associated protein 1 light chain 3 (MAP1-LC3 or LC3). Suggest investigating gene and protein levels. Also Bax/Bcl2 ratio would be informative.
Minor points:
- Please change sacrifice (implies a religious ritual) to “killing of mice”, line 129
- Change “on each” to “in each”, line 131.
- For all graphs, it is better to show all data points superimposed on the bar graphs.
- I struggled to find the definition of CSA, line 190. Please define. It would help to define again in results section, line 308.
- Apoptosis spelling needs fixing, line 548.
Author Response
Amendments to the questions rose by the reviewer #2 to the manuscript ID antioxidants-1313710
- This is an interesting study showing the involvement of the NLRP3 inflammasome in mediating age-relate changes in the cardiac muscle, through the study of the lack of NLRP3. The study has the potential to add useful information to the field if the authors are able to answer some of the below issues. Much is based on observational data and would benefit from more biochemical and functional studies as I highlight below.
Major points:
- Can HW/Tibia length also be reported? This is important as body weight changes with age. If tibia length not available, this should be mentioned as a limitation. This is also important since there appears to be conflicting reports in the literature with one report showing an increase (ref 34) and another showing a reduction (ref 35) in old WT mice.
We are grateful to the reviewer regarding this comment. Concerning normalization of the heart weight to tibia length, unfortunately, we couldn’t get these data because we have not further animals of the old-aged group.
This issue was described in the end of the manuscript under Limitation Section page 17.
- What about Il-1b and IL-18 mRNA? Also it would be interesting to know what happens at the protein level with respect to pro-IL1b and Pro-IL18 and the secreted forms. I suggest Western blots to detect both species. Gasdermin D should also be measured.
We agree with the reviewer concerning the importance of studying the protein content of pro-IL1b and Pro-IL18, and gene expression of Il-1b and IL-18; but unfortunately, we couldn’t analyze these parameters together with Gasdermin D because we have not futther samples of the old-aged group and treated ones.
- Results in aging mitochondria are really interesting. It would be interesting to probe the biochemistry and functional changes behind the observed structural changes. How is OxPhos affected? Suggest performing Seahorse or Oroboros-type experiments to address this. Expression of genes like Opa1 would add to the story. Are ROS levels affected in the aging heart and does NLRP3-/- or melatonin restore this?
We highly appreciated that comment. Actually, we isolated mitochondria and performed respiratory measurements with Seahorse (data not published). Surprisingly, we found no significant differences in OXPHOS among the experimental groups. Taking into account the severe damage observed in the histological and ultrastructure images of cardiomyocytes with aging, and the protective effect of melatonin and NLRP3 absence in the morphometrical parameters, we concluded that respiratory data were not credible. A possible explanation could be the method used to prepare mitochondria, because it has been reported that during the isolation of mitochondria from tissue, most of the impaired mitochondria are broken or lost during the procedures. Therefore, the resultant pool contains relatively well-coupled mitochondria. This phenomenon has been previously observed in mitochondria isolated from liver of septic mice [PMID: 25498899]. We are performing comparative protocols to evaluate the effect of the sample (isolated mitochondria vs. permeabilized fibers from heart) and respirometry technique (Seahorse vs Oroboros) on OXPHOS during aging. The results obtained belong to the project COST Action CA15203 MITOEAGLE and are beyond the aim of this article.
In our previous publication (PMID: 33260800), we assessed age-related impairments in mitochondrial dynamics in the same experimental groups. Specifically, we measured the protein content of Opa1, as well as Mfn2 and Drp1, to evaluate mitochondrial fusion and fission. We referred to our recent published western blot data about age-related impairments of mitochondrial dynamics in the Discussion section, pages 14 and 15, lines 678-728.
Unfortunately, we couldn’t investigate expression of mitochondrial dynamic genes because of lack of samples of the old-aged group and treated ones.
- Autophagy results also interesting but again more is needed to confirm this. Some genes worth looking at include: Beclin1, autophagy related gene 5 (Atg5), and microtubule-associated protein 1 light chain 3 (MAP1-LC3 or LC3). Suggest investigating gene and protein levels. Also Bax/Bcl2 ratio would be informative.
In our recent published article (PMID: 33260800), we performed western blot to evaluate the ratio LC3II/LC3I, which is used as a marker of autophagosomes formation. Bax/Bcl2 ratio, among other parameters related to apoptosis, was also addressed in the same publication. The findings of the current work are a complementary data to analyze the morphological markers of age-related cardiac sarcopenia, as well as to determine the impact of NLRP3 depletion and melatonin therapy on cardiac architecture during aging.
We referred to our recent published western blot data about age-related changes of cardiac autophagy “LC3II/LC3I), and apoptosis (Bax/Bcl2 ratio) in the Discussion section page 15, first and second paragraphs.
Minor points:
- Please change sacrifice (implies a religious ritual) to “killing of mice”, line 129
Revised and corrected.
- Change “on each” to “in each”, line 131.
Revised and corrected.
- For all graphs, it is better to show all data points superimposed on the bar graphs.
We agree with the reviewer concerning this issue; however, after showing all data points on the bar graphs, we found more preferable to use graph bars. The high number of cardiomyocytes analyzed morphometrically, and number of RT-PCR experiments performed made the graph bars more intuitive, understandable.
- I struggled to find the definition of CSA, line 190. Please define. It would help to define again in results section, line 308.
Revised and the definition of CSA was included.
- Apoptosis spelling needs fixing, line 548.
Revised and corrected.
Reviewer 3 Report
This study by Sayed R et al, is an elegant, well-planned study demonstrating the beginning of morphological differences in aging hearts, with cardiac hypertrophy, increased expression of β-MHC, appearance of small necrotic fibers, decline of cadiomyocyte number and increased apoptotic nuclei ratio. These were accompanied by excessive collagen deposition, higher gene expressions of IL-1α, IL-6 and TNFα. Interestingly, genetic ablation of cardiac NLRP3 showed less detectable age-related changes and Oral melatonin therapy preserved the normal cardiomyocytes structure and restored cardiomyocytes number. This study is an extension of a previous study by the authors.
While the study was well-designed and presented, there are a few gaps in the data that can be addressed.
Major Comments:
1) A strong indication of cardiac aging is also a decrease in diastolic function and diastolic filling. It would strengthen the data to include some functional ECHO data.
2) Since NLRP3 is a protein of interest, the gene and protein expression of IL-1b, caspase-1 should be included.
3) In addition to morphological analysis of fibrosis, please include gene expression of key fibrotic markers (CTGF, TGF-b).
4) A recent study looking at cardiac aging by Fabiola Marin-Aguilar was published in Aging Cell 2020 which demonstrated the role of insulin regulation. Please acknowledge this study in either the introduction or the discussion.
5) A more elaborate discussion is needed as to why melatonin was more effective in certain parameters in the NLRP3-/- mice. In particular the B-MHC data is so pronounced with melatonin treatment.
Minor Comments:
1) Figure 3 is labelled as Figure 2.
2) Line 67 : "Accompanied".
3) Please rephrase line 68.
Author Response
Amendments to the questions rose by the reviewer #3 to the manuscript ID antioxidants-1313710
- This study by Sayed R et al, is an elegant, well-planned study demonstrating the beginning of morphological differences in aging hearts, with cardiac hypertrophy, increased expression of β-MHC, appearance of small necrotic fibers, decline of cardiomyocyte number and increased apoptotic nuclei ratio. These were accompanied by excessive collagen deposition, higher gene expressions of IL-1α, IL-6 and TNFα. Interestingly, genetic ablation of cardiac NLRP3 showed less detectable age-related changes and Oral melatonin therapy preserved the normal cardiomyocytes structure and restored cardiomyocytes number. This study is an extension of a previous study by the authors.
- While the study was well-designed and presented, there are a few gaps in the data that can be addressed.
Major Comments:
1) A strong indication of cardiac aging is also a decrease in diastolic function and diastolic filling. It would strengthen the data to include some functional ECHO data.
We are agreeing with the reviewer regarding including functional ECHO data for demonstrating functional decline during aging; however, unfortunately we didn’t have experimental animals for such analyses, especially the old-aged groups. Moreover, we couldn’t get these data using images of MRI included here in the study.
2) Since NLRP3 is a protein of interest, the gene and protein expression of IL-1b, caspase-1 should be included.
We agree with the reviewer concerning the importance of studying the protein content and gene expression IL-1b and caspase-1; but unfortunately, we couldn’t analyze these parameters because of lack samples of the old-aged group and treated ones.
3) In addition to morphological analysis of fibrosis, please include gene expression of key fibrotic markers (CTGF, TGF-b).
We agree with the reviewer concerning this issue that will confirm our morphometrical findings of collagen content; but unfortunately, we couldn’t analyze these parameters together with because of lack samples of the old-aged group and treated ones.
4) A recent study looking at cardiac aging by Fabiola Marin-Aguilar was published in Aging Cell 2020 which demonstrated the role of insulin regulation. Please acknowledge this study in either the introduction or the discussion.
The beneficial effect of NLRP3 inhibition in improvement of autophagy quality during cardiac aging and protecting mice against age-dependent induction of insulin sensitivity was included in the discussion section page 15, lines 742-744.
5) A more elaborate discussion is needed as to why melatonin was more effective in certain parameters in the NLRP3-/- mice. In particular the B-MHC data is so pronounced with melatonin treatment.
A more detailed discussion regarding the higher effectiveness of melatonin in some parameters of mutant mice has been added in the Discussion section pages 15-16, lines 773-823. Furthermore, a statement about the importance of carrying out more studies to elucidate the exact mechanism of this phenomenon was included in the Conclusion section page 17, last sentence.
Minor Comments:
1) Figure 3 is labelled as Figure 2.
Revised and corrected.
2) Line 67: "Accompanied".
Revised and corrected.
3) Please rephrase line 68.
The line was revised and rephrased.
Round 2
Reviewer 2 Report
All my questions are adequately addressed, thank you.
Reviewer 3 Report
It is a pity some of the experiments could not be performed due to the time course of experiments and limited tissues available but it would be good to bear in mind for future experiments to reserve some tissues.